# Clostridium butyricum and Clostridium tyrobutyricum: angel or devil for necrotizing enterocolitis?

Ruizhi Tao,[1] Gangfan Zong,[1] Yehua Pan,[1] Hongxing Li,[2] Peng Cheng,[1] Rui Deng,[1] Wenxing Chen,[1,3] Aiyun Wang,[1,3] Shishan Xia,[4] Weibing Tang,[2] Yin Lu,[1,3] Zhonghong Wei[1,2,3]

**ABSTRACT**  Necrotizing enterocolitis (NEC) is a challenging gastrointestinal disease that disproportionately affects premature neonates, with high incidence and case-fatality rates. Despite extensive research efforts, the pathogenesis and mechanisms of NEC remain unclear, making it difficult to effectively eradicate. However, it has been established that dysbiosis of gut microbes occurs before the onset of NEC, providing compelling evidence for the potential use of probiotic therapy. As such, we have focused our attention on two probiotics in particular: *Clostridium butyricum* and *Clostridium tyrobutyricum*, especially in light of recent breakthroughs that have linked several *Clostridia* species with NEC. To determine whether *C. butyricum* and *C. tyrobutyricum* are pathogenic or probiotic, we conducted a comparison of the phenotypic traits of NEC mice treated with each bacterium. Our results confirm that treatment with *C. tyrobutyricum* restores intestinal barrier integrity and alleviates inflammatory immune responses associated with NEC. In contrast, treatment with *C. butyricum* exacerbates intestinal barrier damage and promotes immune disorder, including increased numbers of macrophages, monocytes, and neutrophils in the intestinal lamina propria. Further analysis of the gut microbiome suggests that the positive effects of *C. tyrobutyricum* treatment are associated with an increase in *Akkermansia muciniphila*, while *C. butyricum* treatment decreases the level of *A. muciniphila*, which accounts for its negative effect on NEC. This study sheds light on the fact that treatment with *C. tyrobutyricum*, but not *C. butyricum*, has the potential to protect against NEC development. The opposite effects of these two probiotics on NEC may result from their different modulation of the level of *A. muciniphila*, a gut microbe that is closely associated with intestinal homeostasis. In summary, by improving the abundance of *A. muciniphila* to alleviate intestinal inflammation and enhance intestinal barrier integrity, supplementation with *C. tyrobutyricum* may become a promising therapy for NEC.

**IMPORTANCE**  This study sheds light on that treatment with *Clostridium tyrobutyricum* but not *Clostridium butyricum* is entitled to protect against necrotizing enterocolitis (NEC) development potentially. The mechanisms behind the opposite effect on NEC may result in different modulation on the level of *Akkermansia muciniphila*, which is deeply associated with intestinal homoeostasis. Briefly, through improving the abundance of *A. muciniphila* to alleviate intestinal inflammation and enhance intestinal barrier integrity, *C. tyrobutyricum* supplement may become a promising therapy for NEC.

**KEYWORDS**  necrotizing enterocolitis, *Clostridium butyricum*, *Clostridium tyrobutyricum*, intestinal barrier

Necrotizing enterocolitis (NEC) is now recognized as the most severe and common acquired gastrointestinal disease in premature infants (1). Up to 12% of premature infants with an extremely low birth weight (VLBW <1500 g) are at higher risk of

Address correspondence to Zhonghong Wei, wzh1225@njucm.edu.cn, Yin Lu, luyingreen@njucm.edu.cn, or Weibing Tang, twbcn@njmu.edu.cn.

Ruizhi Tao and Gangfan Zong contributed equally to this article. Author order was chosen based on draw of straws.

The authors declare no conflict of interest.

See the funding table on p. 16.

*A conflict of interest between the corresponding author and the self-disclosed reviewer was brought to the journal's attention at the proof stage. The handling editor, senior editor, and editor in chief independently verified the robustness of the review, and this paper proceeded to publication.*

developing NEC (2). The development of NEC is attributed not only to antenatal risk factors such as intrauterine inflammation, infection, and preeclampsia (3) but also to postnatal risk factors including prematurity, sepsis, formula feeding, and gut microbiota dysbiosis (4). However, due to inadequate understanding of the complex and multifactorial pathogenesis and mechanisms of NEC, the mortality of NEC neonates remains high, fluctuating between 20% and 30% even after surgery (5). Despite the complex pathogenesis of NEC, a wealth of literature indicates that an imbalance of gut microbiota plays a critical role in the development of the disease, often preceding its onset (6). Early bacterial colonization is also crucial for the formation of intestinal barrier integrity and systemic immune function in infants. When the chemical and physical barriers of the intestine become aberrant, bacterial invasion can occur, leading to perforation, necrosis, and systemic inflammatory reactions (7). Therefore, gut dysbiosis is a core risk factor to be reckoned with during the development of NEC, and probiotic therapy is emerging as an increasingly important approach to address this issue (8, 9). Metagenomic analysis has revealed an increased relative abundance of the phylum *Proteobacteria* and a decreased relative abundance of the phyla *Firmicutes* and *Bacteroidetes* in infants with NEC (10). However, whether the genus *Clostridium* is positively or negatively associated with NEC remains unclear, posing a new challenge for the development of NEC therapeutic strategies. Some studies have suggested that specific strains of *Clostridium*, such as *C. butyricum* (11), *Clostridium perfringens* (12), and *Clostridium neonatale* (13), are positively associated with NEC. However, one study reported a decreased abundance of *Clostridia* with the development of NEC (14). Therefore, we focused our attention on two probiotics of *Clostridium*: *C. butyricum* and *C. tyrobutyricum*, both of which are butyric acid-producing bacteria and have been applied to treat gastroenteritis clinically or experimentally (15–17). To determine whether these *Clostridia* are pathogenic or probiotic for NEC, we compared the phenotypic traits, intestinal barrier integrity, and inflammatory immune response of NEC in a mouse model treated with the two *Clostridia*. Our results reveal that *C. tyrobutyricum* alleviates the symptoms of NEC, while *C. butyricum* promotes the development of NEC. Furthermore, 16S rDNA analysis revealed that *C. tyrobutyricum* treatment enhanced the abundance of *A. muciniphila*, while *C. butyricum* treatment weakened it, indicating the potential for interspecific competition and colonization resistance between these bacteria. Overall, our findings provide new insights into the screening of potential probiotics for clinicians to treat NEC with multiple strategies and highlight the significance of early bacterial colonization for infants, as well as the potential mechanisms of interspecific competition among these bacteria.

## MATERIALS AND METHODS

### Bacterial strains and culture conditions

#### Human sample collection

Human samples, including colons and feces, were collected from the Jiangning Affiliated Hospital of Nanjing Medical University and Nanjing Children's Hospital, respectively, in sterile surgery rooms. Colon samples were transferred using germ-free 50-mL Eppendorf tubes containing sterile RPMI Medium 1640, and feces samples were transferred using germ-free collection tubes. All samples were transported under dry ice conditions for further study. The information for all human samples was provided in Table S1.

*Clostridium butyricum* (ATCC 19398) and *Clostridium tyrobutyricum* (ATCC 25755) were purchased from American Type Culture Collection (ATCC, Manassas, USA). *Akkermansia muciniphila* (DSM 22959) was purchased from Deutsche Sammlung von Mikroorganismen und Zellkulturen (DSMZ, Braunschweig, Germany). Both *Clostridia* were cultured in modified reinforced clostridium medium (RCM) broth (M1285-01, ELITE-MEDIA) at 37°C and *A. muciniphila* (DSM 22959) was cultured in brain heart infusion (BHI) broth (Hopebio, HB8297-5) with 0.1% mucin (SIGMA, M2378-100G) under aerobic conditions.

## Mouse model of necrotizing enterocolitis with or without treatment

Pregnant C57/BL6 mice were purchased from Jiangsu Jicuiyaokang Biotechnology Co., LTD. When newborn mice were 1 wk old, mice were randomly divided into different groups. Control mice were fed with breast milk without separated from their dams. Necrotizing enterocolitis (NEC) mice were co-housed in a neonatal incubator at 32-35℃ and 60-70% humidity. They were hand-fed with formula milk (40 µL/g), consisted of 15 g Similac Advance infant formula (Abbott Nutrition) and 10 g Esbilac canine milk replacer (PetAg) in 75 mL water with or without bacteria (12.5 µL stool slurry per milliliter or $10^9$ colony-forming units per milliliter) six times a day. Meanwhile, they were stressed with hypoxia (5% $O_2$, 95% $N_2$) for 10 min, followed by cold stress (placement in a refrigerator at 4℃) for 5 min later twice a day. Mice were humanely sacrificed after 1 wk and the intestines were collected immediately for further experiments. Briefly groups were as follows: control group, breast-fed only; NEC group, formula-fed, hypoxia, and cold stress; feces group, formula-fed with bacterial slurry, hypoxia, and cold stress; *C. butyricum* group, formula-fed with *C. butyricum*, hypoxia, and cold stress; *C. tyrobutyricum* group, formula-fed with *C. tyrobutyricum*, hypoxia, and cold stress.

## Histological analysis

Mouse intestinal tissue sections were soaked in 4% Carnoy's fixative for 24 h and paraffin-embedded. Then, hematoxylin-eosin staining (Leagene, DH0006) and Alcian blue staining (Leagene, 0041) were performed on paraffin sections according to manufacturer's protocol. All histological evaluation of the ileum and colon were conducted and scored in a blinded manner. The total score obtained was statistically analyzed. Alcian blue staining was used for quantifying the coverage of mucus layer.

The ileum epithelial histological score was performed as follows (18)：normal (score, 0); mild (score, 1), separation of the villus core, without other abnormalities; moderate (score, 2), villus core separation, submucosal edema, and epithelial sloughing; severe (score, 3), denudation of epithelium with loss of villi, full thickness necrosis, or perforation.

The colon epithelial histological score was performed as follows (19)：normal (score, 0); excessive proliferation, abnormal crypt morphology, and goblet cell deletion (score, 1); moderate recess loss (10-50%) (score, 2); severe crypt loss (50-90%) (score, 3); complete absence of crypt (score, 4); small- and medium-sized ulcer (ulcer surface <10 recess length), (score, 5); large ulcer (length of ulcer surface ≥10 crypts) (score, 6).

## Real-time quantitative PCR analysis

Intestine tissues were homogenized using a Freeze grinder (Shanghai Jingxin) immediately after being lysed in TRIzol (Thermo Fisher, USA). RNA was isolated by chloroform-isopropanol method and cDNA was synthesized using an iScript cDNA Synthesis Kit (Vazyme, China). cDNA was synthesized from 500 ng total RNA using HiscriptII QRT SuperMix (Vazyme, China). Real-time PCR was performed using ChamQ SYBR qPCR Master Mix (Low ROX Premixed) (Vazyme, China) and detected by ABI 7500 system (Applied Biosystems, CA, USA). Primers were used as follows: *Muc2* F, 5′-TCCAGAAAGAA GCCAGATCC-3′; *Muc2* R, 5′-ACACTGCTCACAGTCGTTGG-3′; *Muc5ac* F, 5′-ACATTTCCCCATG CTCCACAGC-3′; *Muc5ac* R, 5′-GTGGTGGTATTAGACTCCTGG-3′; *Tff1* F, 5′-CCCGGGAGAGGAT AAATTGT-3′; *Tff1* R, 5′-GCCAGTTCTCTCAGGATGGA-3′; *Tff3* F, 5′-TAATGCTGTTGGTGGTCCTG -3′; *Tff3* R, 5′-CAGCCACGGTTGTTACACTG-3′; *Tlr4* F, 5′-CATGGAACACATGGCTGCTAA-3′; *Tlr4* R, 5′-GTAATTCATACCCCTGGAAAGG-3′; *Myd88* F, 5′-CCACCTGTAAAGGCTTCTCG-3′; *Myd88* R, 5′-CTAGAGCTGCTGGCCTTGTT-3′; *NfκB* F, 5′-AACCAATCTTACCGCTCCCG-3′; *NfκB* R, 5′-C GAGTAGCCGCCGTAATAGG-3′.

## Isolation of the lamina propria mononuclear cells (LPMCs) from ileum murine and colon

Fresh ileum and colon tissues were collected when mice were sacrificed. After being washed with cold PBS, tissues were cut into 1.5 cm and incubated in 1 mM dithiothreitol (DTT, Sigma-Aldrich, D9779) and 10 mM HEPES solution (Sigma-Aldrich, 375368) for 10 min at 37°C. After vigorous shaking and washing with PBS, we used 30 mM ethylenediaminetetraacetic acid (EDTA, Sigma-Aldrich, 798681) and 10 mM HEPES solution to incubate tissues twice. Then tissues were incubated in RPMI Medium 1640 (Gibco, 31800-022) with 0.2 mg/mL collagenase I (Sigma-Aldrich, C2674) and 0.15 mg/mL DNase (Sigma-Aldrich, AMPD1) to digest tissues for 90 min at 37°C. After a vigorous stirrer, the cells were purified by using a 70-μm filter and were collected by centrifugation for 5 min at 500$g$. A discontinuous Percoll (Cytiva, 17089102) gradient (80%/40%) was then used to separate mononuclear cells through centrifugation for 30 min at 1,000$g$ without break. The LPMCs were collected at the interface between two Percoll gradients for further flow cytometry.

## Flow cytometry analysis

The cell was labeled with FITC anti-mouse CD45 Antibody (BioLegend, No. 103107), APC anti-mouse/human CD11b (BioLegend, No. 101211), PE/cyanine seven anti-mouse F4/80 (BioLegend, No. 123113), PE anti-mouse Ly-6C (BioLegend, No. 128007), and PerCP/cyanine 5.5 anti-mouse Ly-6G (BioLegend, No. 127616) for 20 min at room temperature. Flow cytometry analysis was performed on a CytoFLEX (Beckman Coulter), and results were analyzed by FlowJo software version 10.

## Immunofluorescence imaging

Intestinal samples were fixed overnight in 4% Carnoy's fixative. Samples were then dehydrated in gradient ethanol (100%–95%–90%–75%–50%) and Xylene solution for at least 4 h. Then 4-μm paraffin sections were rehydrated, blocked with 3% BSA solution, and stained with antibodies overnight as follows: ZO-1 (Proteintech, 21773-AP, 1:2,000), E-cadherin (Affinity, AF0131, 1:200), and claudin-1 (Affinity, DF6919, 1:200). Tissue sections were then incubated with the appropriate fluorophore-conjugated secondary antibody. Before imaging, nuclei were counterstained with DAPI or Hoechst. Fluorescence analysis was performed on a Leica SP8 Fluorescence microscope.

## Fluorescence *in situ* hybridization

Ileum tissues were fixed in 4% Carnoy's fixative and embedded in paraffin, as previously described. The tissues were then sectioned to a thickness of 4 μm and hybridized with a universal bacterial probe targeting the 16S rRNA gene (EUB338 probe: 5′-GCTGCC TCCCGTAGGAGT-3′). The samples were stained overnight at 4°C with mucin2 (Santa, sc-515032, 1:300) and Hoechst for 10 min at room temperature. Fluorescence analysis was performed using a Leica SP8 fluorescence microscope. The percentage of bacteria was quantified per square millimeter of ileum in each group.

## Western blot

Tissues were lysed in lysis buffer (P0013B, Beyotime) containing protease inhibitor cocktail (ST506, Beyotime) and phosphatase inhibitor cocktail (GB-0032, KeyGEN) using a low-temperature freeze grinder. Proteins were denatured and separated by SDS-PAGE, then transferred onto PVDF membranes (Millipore, USA). The samples were probed with antibodies overnight at 4°C and labeled with goat anti-rabbit IgG (H+L) HRP antibodies (ZJ2020-R, Bioworld). The samples were imaged using a BIORAD imaging system (chemiDOCTMXRS, Bio-Rad). The primary antibodies used were as follows: ZO-1 (Proteintech, 21773-AP, 1:5,000), E-cadherin (G-10) (Santa Clara, sc-8426,1:500), claudin-1 (Affinity, DF6919, 1:1000), TLR4 (Santa, sc-293072, 1:500), MyD88 (CST, D80F5, 1:1,000),

NF-κB (Santa Clara, sc-8008, 1:500), Phospho-NF-κB (Affinity, AF2006, 1:1,000), and anti-GAPDH (MA5-15738, ABclonal, 1:10,000).

## 16s ribosomal RNA gene sequencing

Total genomic DNA was extracted from samples using Hexadecyltrimethy Ammonium Bromide (CTAB). 16s rDNA genes were amplified using a specific primer with the barcode. All PCR reactions were carried out using Phusion Hot Start II High-Fidelity PCR Master Mix (Thermo Fisher, USA). The universal bacterial 16s rRNA gene amplicon PCR primers were used: the forward primer was 5′-CCTACGGGNGGCWGCAG-3′, and the reverse primer was 5′-GACTACHVGGGTATCTAATCC-3′. Then, the DNA monitored on 2% agarose gel was purified by means of AMPure XT beads (Beckman Coulter Genomics, Danvers, MA, USA) and quantified by Qubit (Invitrogen, USA). Sequencing libraries were generated using Agilent 2100 bioanalyzer (Agilent, USA) and Illumina (KapaBiosciences, Woburn, MA, USA). Then paired-end sequencing was performed using NovaSeq 6000 sequenator (Illumina, San Diego, CA, USA) and NovaSeq 6000 SP Reagent Kit (500 cycles) (Illumina, 20029137).

## RNA sequencing

Total RNAs of colon were extracted using Trizol Reagent and purified by Ribo-ZeroTM Magnetic Gold Kits (Illumina, MRZG126) to remove rRNA. We collected 3 µg RNAs of each biological sample to construct sequencing library using NEB Next Ultra Directional RNA LibraryPrep Kit for Illumina (NEB, Ispawich, USA). The mRNAs were measured by DEGseq software to find differential expressed genes (DEG) and DEGs were further annotated through NCBI, Uniport, GO, and KEGG database.

## Cell viability assay

Cell viability was determined by the Cell Counting Kit-8 (CCK-8, APExBIO, K1018) assay. We seeded 1000 cells for 96-well plate in each well, respectively. After being treated with different bacteria directly with multiplicity of infection of 100 for 4 h under anaerobic condition, we removed the culture medium containing bacteria instead of 1640 medium with 10% FBS, 1% penicillin/streptomycin, and 100 mg/mL gentamycin for 2 h to kill the extracellular bacteria in wells. Then the medium was replaced by 1640 medium supplemented with 10% FBS and 10% CCK-8 for further analysis.

## Bacterial competition in liquid cultures

The overnight cultures of *C. butyricum*, *C. tyrobutyricum*, and *A. muciniphila* were prepared for competition experiments beforehand. For direct competition, all bacterial cultures were diluted to an initial $OD_{600}$ of 0.1 and bacteria were centrifuged at 2,000 rpm for 5 min. For hot-killed bacterial experiment, *C. butyricum* and *C. tyrobutyricum* cultures were placed in an air oven at 70°C for 30 min to be killed completely. Then *C. butyricum* and *C. tyrobutyricum* were inoculated with *A. muciniphila,* respectively, resuspended in a mixed culture medium (half RCM and half BHI with 0.1% mucin), whereas co-culture of *C. butyricum* + *A. muciniphila* and *C. tyrobutyricum* + *A. muciniphila* were tested with an initial $OD_{600}$ of 0.05 of each strain. For bacterial indirect competition, after the *C. butyricum* and *C. tyrobutyricum* cultures grew overnight at an initial $OD_{600}$ of 0.1in a mixed culture medium, the conditioned medium supernatant was collected by centrifuging at 4,000 rpm for 10 min. Then conditioned mediums were used to culture *A. muciniphila* respectively with an initial $OD_{600}$ of 0.05 of *A. muciniphila*. Bacterial cultures were collected at 0 h, 24 h, 48 h and 72 h of each experiment simultaneously. Total bacterial DNA were extracted using Bacterial DNA Kit (OMEGA, D3350-01). Quantitative PCR on a 7500 Sequence Detector (Applied Biosystems, CA, USA) was used to calculate the number of *C. butyricum, C. tyrobutyricum*, and *A. muciniphila* 16s rDNA gene copies in the co-culture medium. The primers were as follows: *A. muciniphila* F, 5′-GTTCGGAATCAC TGGGCGTA-3′; *A. muciniphila* R, 5′-CGCATTTCACTGCTACACCG-3′.

## Statistical analysis

Statistical significance was determined with *t*-test, one-way ANOVA test, or two-way ANOVA test, as indicated in the figure legends. Error bars indicate SEM on all graphs. Prism 9.0 (Graphpad, La Jolla, CA) was used for all statistical analyses.

## RESULTS

### Gut microbiota is involved in the development of NEC

To investigate whether dysbiosis of gut microbiota is associated with the development of NEC, we compared the gut microbiota composition of NEC infants with that of normal neonates using 16S rDNA sequencing. Although there was no significant difference in alpha diversity (Fig. 1A), NMSD analysis revealed a distinct change in beta diversity of microbiota composition between the NEC group and the control group (Fig. 1B). The differential taxonomic similarity of intestinal bacteria at the phylum level between each group was reflected in *Actinobacteriota*, *Firmicutes*, and *Proteobacteria* (Fig. 1C). Notably, bubble plot analysis also confirmed a significant alteration at the genus level (Fig. 1D). Microbial network analysis also suggested potential interactions between specific groups of gut microbiota (Fig. S1A). The PICRUSt2 predicted results indicate that the distinct microbiome in NEC patients plays a crucial role in the metabolism and biosynthesis of amino acids, including arginine, proline, and histidine. These findings highlight the significant impact of the differentiated microbiome on the host's amino acid processes (Fig. S1B).

To further investigate the underlying mechanism, we used a mouse model to induce NEC, including formula feeding, hypoxia, and cold stress (20), and we treated the mice with bacterial stock (12.5 µL stool slurry in 1 mL formula) cultured from the stool of NEC patients for 1 wk (Fig. 1E). Considering that NEC infants are likely to present with pneumatosis cystoides intestinalis (21), we compared the severity of pathology in each group and calculated the number of pneumatosis cystoides intestinalis (Fig. 1F and G). The results showed that bacterial treatment facilitated the luminal gas trapping of NEC mice. Moreover, ileum and colon tissue damages were more severe after bacterial gavage (Fig. 1H), as judged from H&E staining. Overall, our data indicate that microbiome disorder promotes the development of NEC.

### *C. tyrobutyricum* protected against NEC while *C. butyricum* worsened NEC

Recent studies, including epidemiological studies (22), clinical signs (23), and animal models (24), have demonstrated the involvement of *Clostridia* in the development of NEC. To investigate whether two different *Clostridium* probiotics, *C. tyrobutyricum* and *C. butyricum*, can be used for NEC treatment, we fed NEC mice with mixed formula containing both probiotics (at $10^9$ CFUs/mL), as described previously (20) (Fig. 2A). We found that mice treated with *C. tyrobutyricum* had milder symptoms of pneumatosis cystoides intestinalis, while the symptoms of mice treated with *C. butyricum* became more severe (Fig. 2B and C). Furthermore, H&E staining results verified that *C. tyrobutyricum* alleviated ileum and colon damage, while *C. butyricum* aggravated the damage (Fig. 2D).

These findings suggest that each *Clostridium* strain has unknown mechanisms that impact the development of NEC, resulting in the opposite effects observed with *C. tyrobutyricum* and *C. butyricum*. Overall, the study indicates the potential therapeutic benefits of *C. tyrobutyricum* and the potential harm of *C. butyricum* in the treatment of NEC.

### Intestinal inflammation was alleviated by *C. tyrobutyricum* but aggravated by *C. butyricum*

To investigate the mechanisms underlying the effects of both *Clostridia* species on the progression of NEC, the study focused on alterations at the transcriptional level of the disease itself. RNA sequencing was performed to assess messenger (m)RNA expression

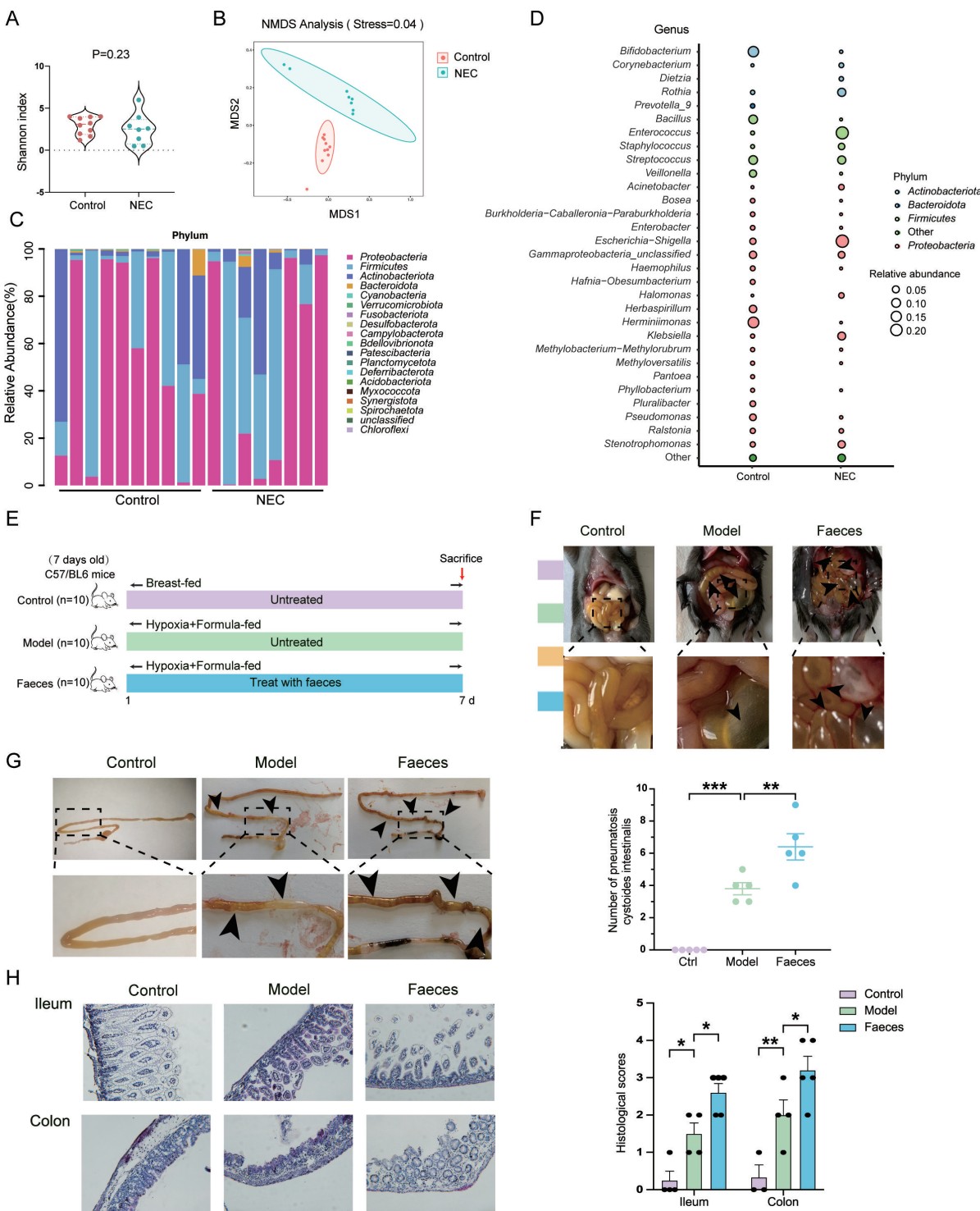

**FIG 1** Gut microbiota is involved in the development of NEC. (A) Shannon index of bacteria in human feces ($n$ = 8–10). (B) NMDS analysis of bacteria in human feces using Bray-Curtis metric distances of beta diversity ($n$ = 8–10). (C and D) Relative abundance of bacteria composition at phylum and genus level ($n$ = 8–10). (E) Experimental design for transplanting faeces from NEC patients to mice under NEC model ($n$ = 10). (F and G) Representative images of enteric cavity and canal at 7 d. The number of pneumatosis cystoides intestinalis marked by arrows were counted and shown as scatter plots ($n$ = 5). (H) H&E staining and histological scores of ileum and colon ($n$ = 3–5). Quantified results were shown as mean ± SEM. $p$-Values were generated by one-way ANOVA with multiple comparisons. *$P$ < 0.05, **$P$ < 0.01,*** $P$ < 0.001.

between NEC and normal neonate colon tissues, taking into consideration that the usual sites of NEC were the distal ileum and proximal colon (25). As expected, there was a significant difference in mRNA expression between the two groups (Fig. 3A). A total of 1970 upregulated mRNAs were identified, and differential inflammatory-related genes were labeled in the volcano plot (Fig. 3B). Furthermore, KEGG pathway enrichment revealed that Toll-like receptor signaling pathway and NF-κB signaling pathway were significantly upregulated (Fig. 3C), indicating the presence of an inflammatory state in neonates with NEC.

To further confirm the activation of intestinal inflammation, we extracted immune cells from the lamina propria layer of the ileum and colon of the mice (26). The number of CD11b⁺ F4/80⁺ macrophages, CD11b⁺ LY6c⁺ monocytes, and CD11b⁺ LY6g⁺ neutrophils was detected using flow cytometry in each group. Notably, treatment with *C. tyrobutyricum* decreased the number of macrophages, monocytes, and neutrophils increased by NEC, while treatment with *C. butyricum* failed to reverse it (Fig. 3D; Fig. S2A). We also detected Th17/Treg cells in four different groups. Our results showed that the number of CD4+RORγt+ (Th17) cells and CD25+FoxP3+ (Treg) cells was significantly altered in both

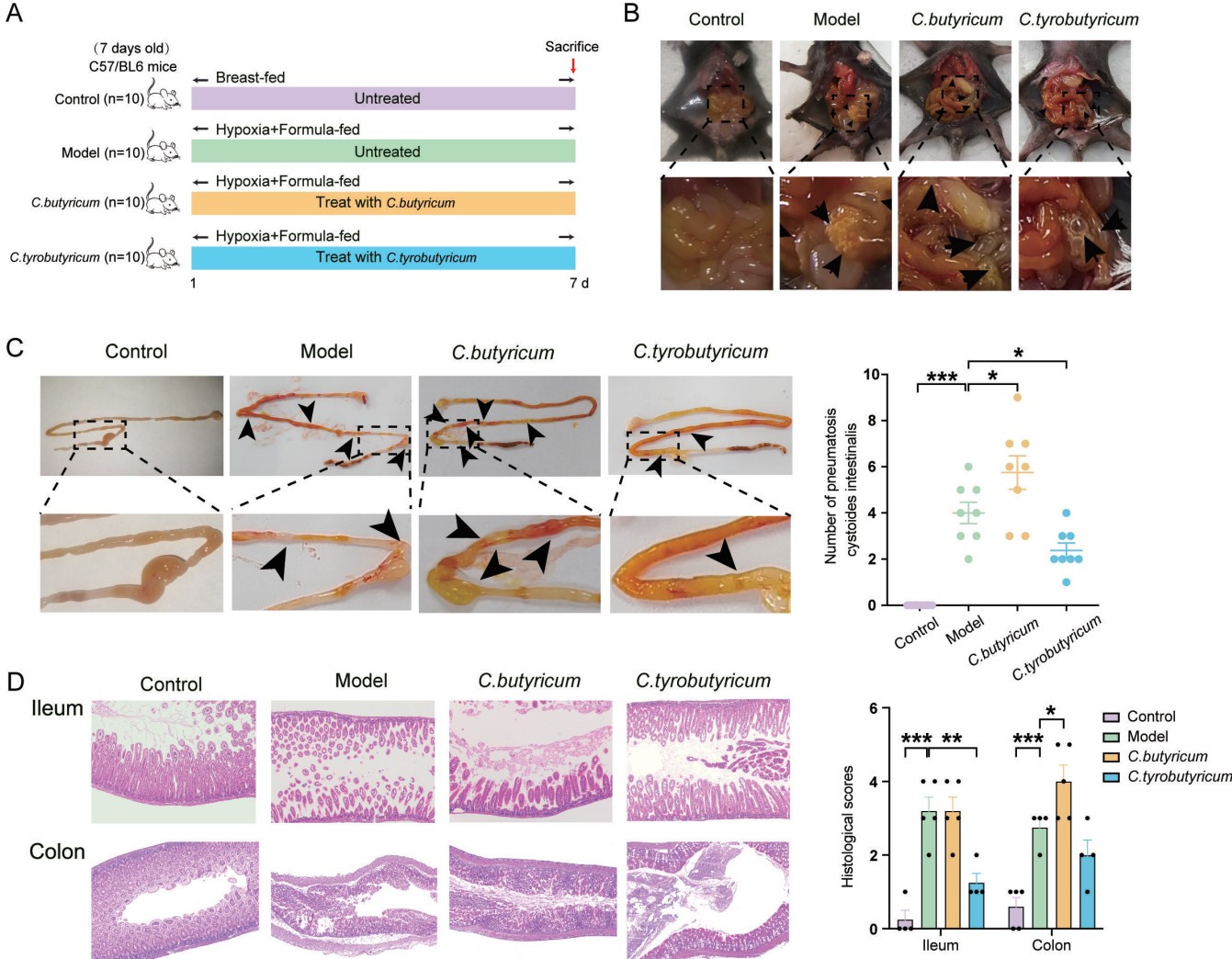

**FIG 2** *C. tyrobutyricum* protected against NEC while *C. butyricum* worsened NEC. (A) Experimental design for treating mice with *C. butyricum* ($1 \times 10^9$ CFU/day) and *C. tyrobutyricum* ($1 \times 10^9$ CFU/day) under NEC model (*n* = 10). (B and C) Representative images of enteric cavity and canal at 7 d. The number of pneumatosis cystoides intestinalis marked by arrows were counted and shown as scatter plots (*n* = 8). (D) H&E staining and histological scores of ileum and colon (*n* = 3–5). Quantified results were shown as mean ± SEM. *P*-values were generated by one-way ANOVA or two-way ANOVA with multiple comparisons. **P* < 0.05, ***P* < 0.01,****P* < 0.001.

the colon and ileum. Specifically, the NEC model disrupted the Th17/Treg cell balance in the ileum and colon. Treatment with *C. tyrobutyricum* reversed the decrease of Th17 cells in NEC model mice and decreased the number of Treg cells, while *C. butyricum* failed to produce the same effect (Fig. S2B and C).

Consistent with the RNA-sequencing data, the expression of genes in intestinal tissues, such as *Tlr4*, *Myd88*, *Nf-κb*, and *Il-1β*, were significantly upregulated and reversed by *C. tyrobutyricum*, but not by *C. butyricum* (Fig. 3E). The protein level of intestinal tissues, including TLR4, MyD88, NF-κB, and phospho-NF-κB, was examined by western blot (WB). Treatment with *C. butyricum* elevated the expression of TLR4, MyD88, and phospho-NF-κB, while TLR4 and phospho-NF-κB were reduced when treated with *C. tyrobutyricum* (Fig. 3F).

Overall, the data suggest that NEC mice experienced severe intestinal inflammation, and the two different *Clostridia* probiotics had opposite effects. *C. tyrobutyricum* alleviated the inflammation through modulating immune cells and the TLR4/NF-κB signaling pathway, while *C. butyricum* aggravated the inflammation.

## Intestinal barrier integrity was protected by *C. tyrobutyricum* but disrupted by *C. butyricum*

RNA-sequencing data were further analyzed to determine the alterations between NEC and normal neonates at the mRNA level in the colon. Gene set enrichment analysis (GSEA) of the differentially expressed genes between NEC and normal neonates detected enrichment of downregulated genes characteristic of tight junctions (Fig. 4A). We labeled the genes related to tight junctions among the 2073 downregulated genes in the volcano plot (Fig. 4B). Similarly, KEGG pathway enrichment emphasized that the tight junction signaling pathway was significantly inhibited, indicating damage to the intestinal barrier integrity in NEC neonates (Fig. 4C).

To investigate the link between intestinal barrier integrity and the opposite effects of two *Clostridia* on the development of NEC, we performed Alcian blue staining to observe the thickness of the mucus layer in the ileum (Fig. 4D). Treatment with *C. tyrobutyricum* significantly increased the thickness of the mucus layer, while treatment with *C. butyricum* significantly reduced it. Furthermore, more bacteria were translocated into the submucous layer in the ileum of NEC mice, and treatment with *C. butyricum* failed to alleviate the permeability of the intestinal barrier, in contrast to the dramatic protection provided by *C. tyrobutyricum* treatment (Fig. 4E). Meanwhile, the expression of the tight junction protein ZO-1 and claudin-1 and the cell adhesion protein E-cadherin in the ileum and colon were reduced in NEC mice through immunofluorescence and western blotting, and *C. tyrobutyricum* treatment remarkably improved the expression of ZO-1 and E-cadherin (Fig. 4F and G; Fig.S3A). We also examined the mRNA levels of *Muc2*, *Muc5ac*, *Tff1*, and *Tff3*, which are associated with mucus barrier integrity (27), in intestinal tissues through real-time PCR (Fig. 4H). The results further confirmed the protective effect of *C. tyrobutyricum* on the mucus barrier. All in all, these data suggest that intestinal barrier integrity is protected by *C. tyrobutyricum* but disrupted by *C. butyricum*.

## The positive effect of *C. tyrobutyricum* and the negative effect of *C. butyricum* on NEC were associated with modulating the level of *A. muciniphila*

We performed bacterial 16s rDNA sequencing on mouse feces to investigate the potential association between gut microbiota dysbiosis and NEC development, as well as the effects of two *Clostridia* on microbiome composition. Although the bacteria Shannon index of the four groups did not show significant changes (Fig. S4A), three-dimensional principal component analysis revealed a distinct separation of microbiota for the control, model, *C. butyricum*, and *C. tyrobutyricum* groups (Fig. 5A). We further investigated the overall bacterial composition among each group by comparing the top 10 relative abundance of bacteria at the phylum level (Fig. 5B). Consistent with a previous study (10), the levels of *Bacteroidetes*, *Proteobacteria*, and *Firmicutes* were significantly altered.

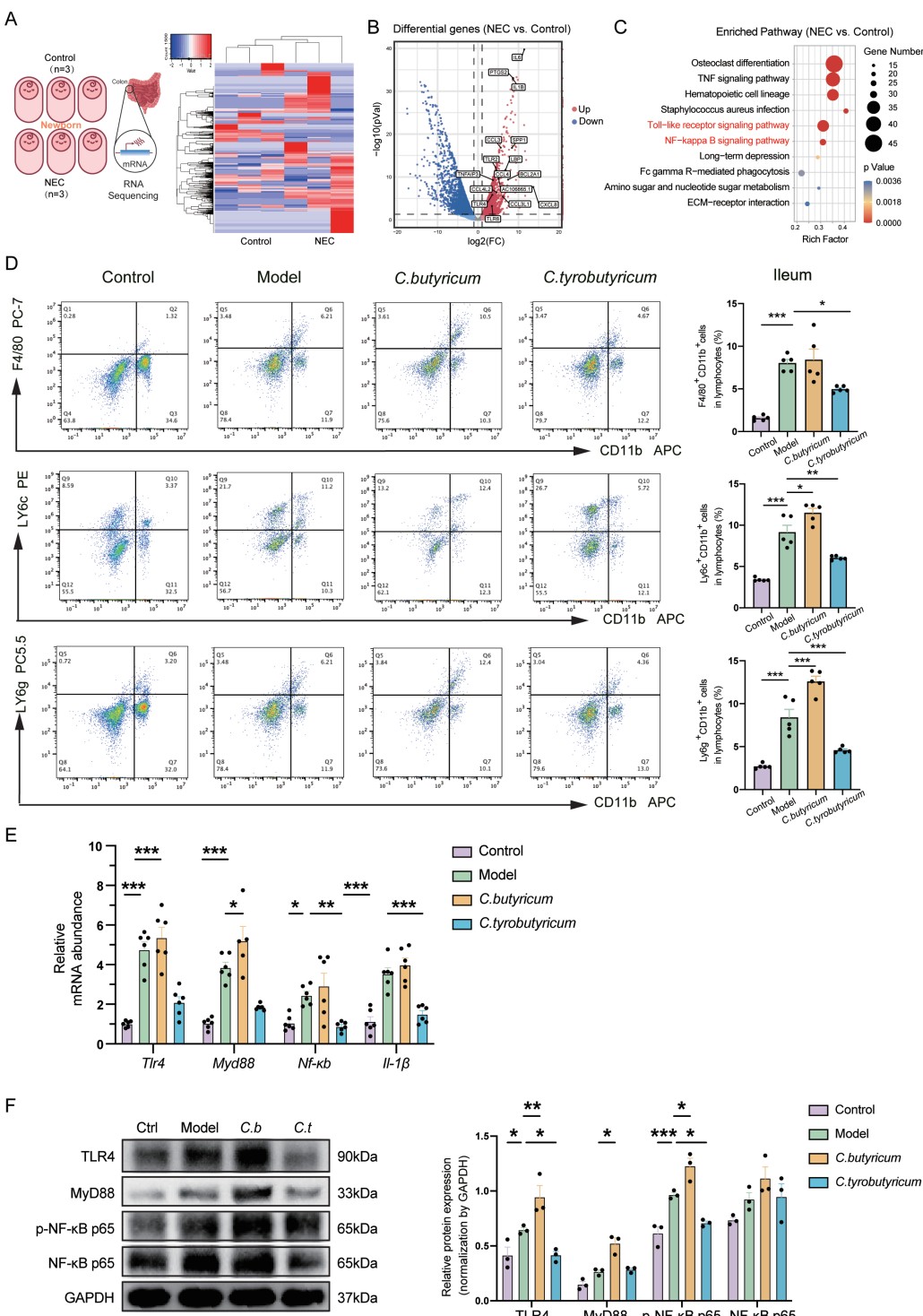

**FIG 3** Intestinal inflammation was alleviated by *C. tyrobutyricum* but aggravated by *C. butyricum*. (A) Schematic diagram demonstrating the workflow of RNA-seq for colons of NEC and normal neonate ($n = 3$). (B) Volcano plot showing the differential genes and labeling some genes upregulated in comparison between NEC and control. (C) Analysis of top 10 upregulated pathways in NEC compared with control. (D) Flow cytometric analysis of CD11b⁺F4/80⁺ (macrophages), CD11b⁺Ly6C⁺ (monocytes), and CD11b⁺Ly6G⁺ (neutrophils) cells in the ileum tissues of mice ($n = 5$). (E) mRNA expressions of *Tlr4*, *Myd88*, *Nf-κb*, and *Il-1β* in mice intestine tissues with *C. butyricum* or *C. tyrobutyricum* treatment ($n = 5$–6). (F) Protein level of TLR4, MyD88, phospho-NF-κB, and NF-κB in intestine tissues ($n = 3$). Quantified results were shown as mean ± SEM. *P*-values were generated by one-way ANOVA or two-way ANOVA with multiple comparisons. *$P < 0.05$, **$P < 0.01$, *** $P < 0.001$.

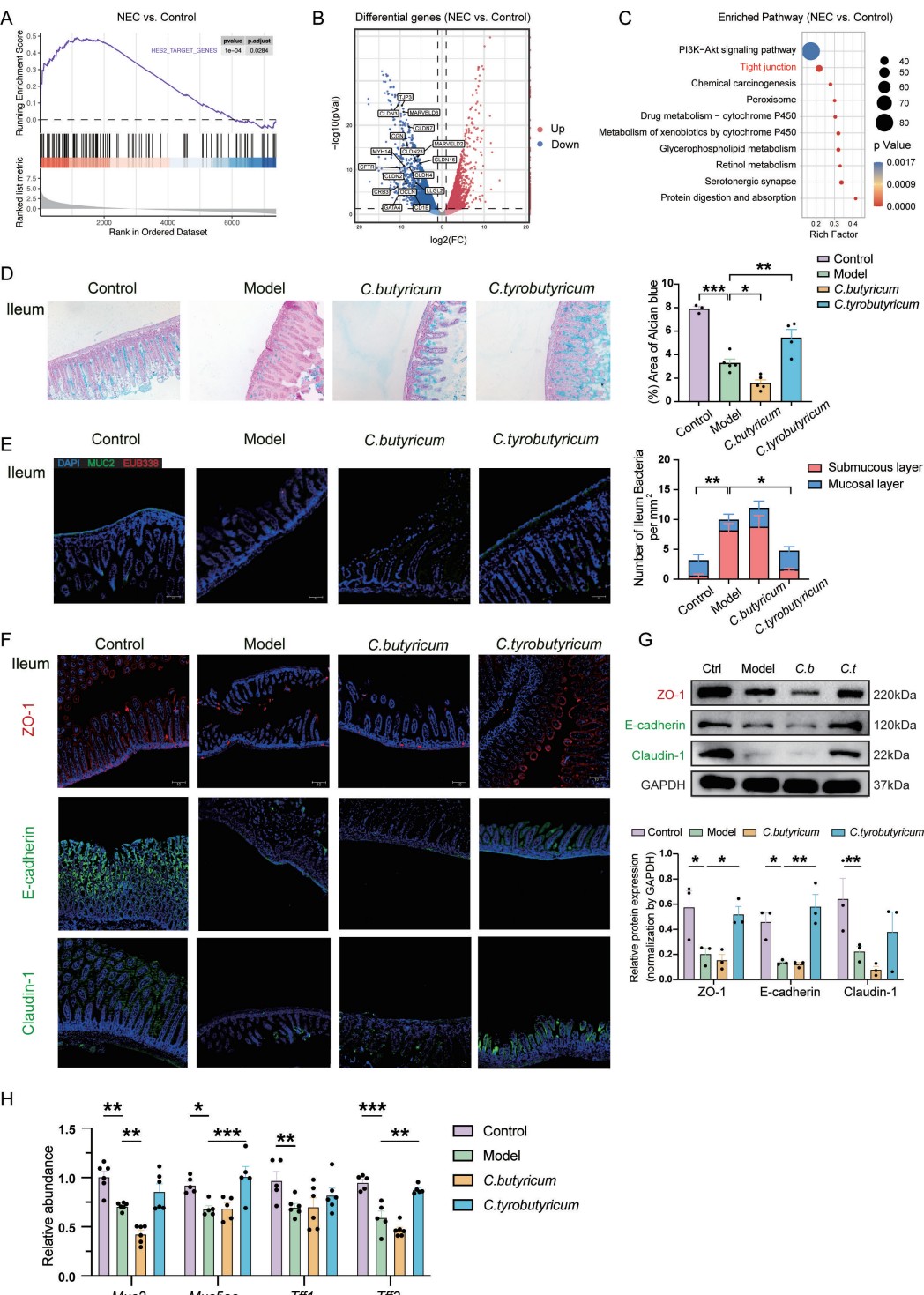

**FIG 4** Intestinal barrier integrity was protected by *C. tyrobutyricum* but disrupted by *C. butyricum*. (A) GESA plot of genes enriched in NEC versus control was performed with MSigDB (C3). (B) Volcano plot showing the differential genes and labeling some genes downregulated in comparison between NEC and control. (C) Analysis of top 10 downregulated pathways in NEC group compared with control group. (D) Representative Alcian blue images of ileum (100×) and its analysis of each group is shown as a histogram (*n* = 3–5). (E) Bacteria were stained with EUB338 (red), MUC-2 (green), and DAPI (blue) by fluorescent *in situ* hybridization (FISH) in each group of ileum (100×), scale bar: 100 μm. (*n* = 5) (F) Immunofluorescence analysis on ZO-1, E-cadherin, and claudin-1 in ileum sections from different groups. Representative images were shown (100×). Scale bar: 100 μm. (G) Protein level of ZO-1, E-cadherin, and claudin-1 in mice intestine sections (*n* = 3). (H) mRNA expressions of *Muc2*, *Muc5ae*, *Tff1*, and *Tff3* in mice intestine tissues (*n* = 5–6). Quantified results were shown as mean ± SEM. *P*-values were generated by one-way ANOVA or two-way ANOVA with multiple comparisons. *$P < 0.05$, **$P < 0.01$, *** $P < 0.001$.

We observed a decrease in *Verrucomicrobia* in the NEC and *C. butyricum* groups and an increase in the *C. tyrobutyricum* group. The analysis of the genus confirmed the decline of *Akkermansia* in the NEC group and the improvement of *Akkermansia* in the *C. tyrobutyricum* group (Fig. 5C; Fig. S4B). The result of the linear discriminant analysis effect size (LEfse) analysis also revealed this phenomenon in mice feces (Fig. 5D). Consistently, the abundance of *Akkermansia* significantly decreased in neonate samples (Fig. 5E).

To date, *A. muciniphila* is a well-known probiotic species in the genus *Akkermansia* that has been extensively studied and demonstrated to have beneficial effects on gut health. As a mucus-consuming bacterium in the intestine, the reduction of *A. muciniphila* is strongly associated with multiple diseases (28), and proper supplementation of *A. muciniphila* has been shown to improve intestinal barrier integrity and host immunity (29). Therefore, we measured the relative abundance of *A. muciniphila* using quantitative PCR in different mouse groups and found the same variation trend in mice feces (Fig. 5F). Overall, our findings suggest that, while *C. butyricum* treatment worsened the decline of *A. muciniphila* caused by the NEC model, *C. tyrobutyricum* treatment increased the abundance of *A. muciniphila* in mice feces.

## Interspecific competition provided a fitness advantage to *C. butyricum* over *A. muciniphila*

To investigate the reason behind this observation, we examined whether interspecific colonization resistance between the two *Clostridia* and *A. muciniphila* could account for the variation of *A. muciniphila*. We prepared overnight liquid cultures of each strain for bacterial competition (30). We compared the growth rates of *A. muciniphila* directly with *C. butyricum* and *C. tyrobutyricum* under nutrient-limited conditions using absolute quantitative PCR (Fig. 6A). The results showed that *A. muciniphila* had lower growth rates when co-cultured with *C. butyricum* than when grown alone, but the yields did not change significantly when co-cultured with *C. tyrobutyricum*. We also found that hot-killed *C. butyricum* suppressed the growth of *A. muciniphila* (Fig. 6B), but the conditioned medium of *C. butyricum* had no effect on the growth of *A. muciniphila* (Fig. 6C). These findings suggest that the colonization resistance of *C. butyricum* to *A. muciniphila* is rooted in the bacterial structure itself and that the fermentation products or metabolites of *C. butyricum* and *C. tyrobutyricum* do not act as stimulatory molecules. Taken together, our data indicate that interspecific competition provided a fitness advantage to *C. butyricum* over *A. muciniphila*.

## Conclusion

NEC has become a serious health threat in infants, particularly premature neonates. Despite efforts to understand its pathogenesis and underlying mechanisms, there is still an urgent need for more effective and safe therapeutic approaches (31). The gut microbiome plays a crucial role in the healthy growth of infants, including immune and intestinal barrier development, and recent insights suggest that gut microbiome dysbiosis occurs before the onset of NEC (32). Our 16S rDNA data and experimental design confirmed the involvement of gut microbiota in NEC development. Probiotic therapy has been shown to be beneficial for restoring intestinal homeostasis and reducing the incidence of NEC (33, 34). However, not all probiotic products are suitable for NEC treatment, and there is still controversy regarding the effect of *Clostridia* on NEC development, with *C. butyricum*, a butyric acid-producing probiotic, being implicated in NEC (11).

To address this issue, we chose two *Clostridium* probiotics, *C. butyricum* and *C. tyrobutyricum*, to determine whether they are suitable for treating NEC. We established an NEC mouse model and fed the mice with exclusive formula mixed with either *C. butyricum* or *C. tyrobutyricum*. Surprisingly, *C. tyrobutyricum* treatment alleviated the severity of the NEC model, while *C. butyricum* treatment worsened the condition. Our RNA-sequencing results showed that NEC neonates had more intense inflammatory signaling in their intestines (35) and lower intestinal barrier integrity (36). *C.*

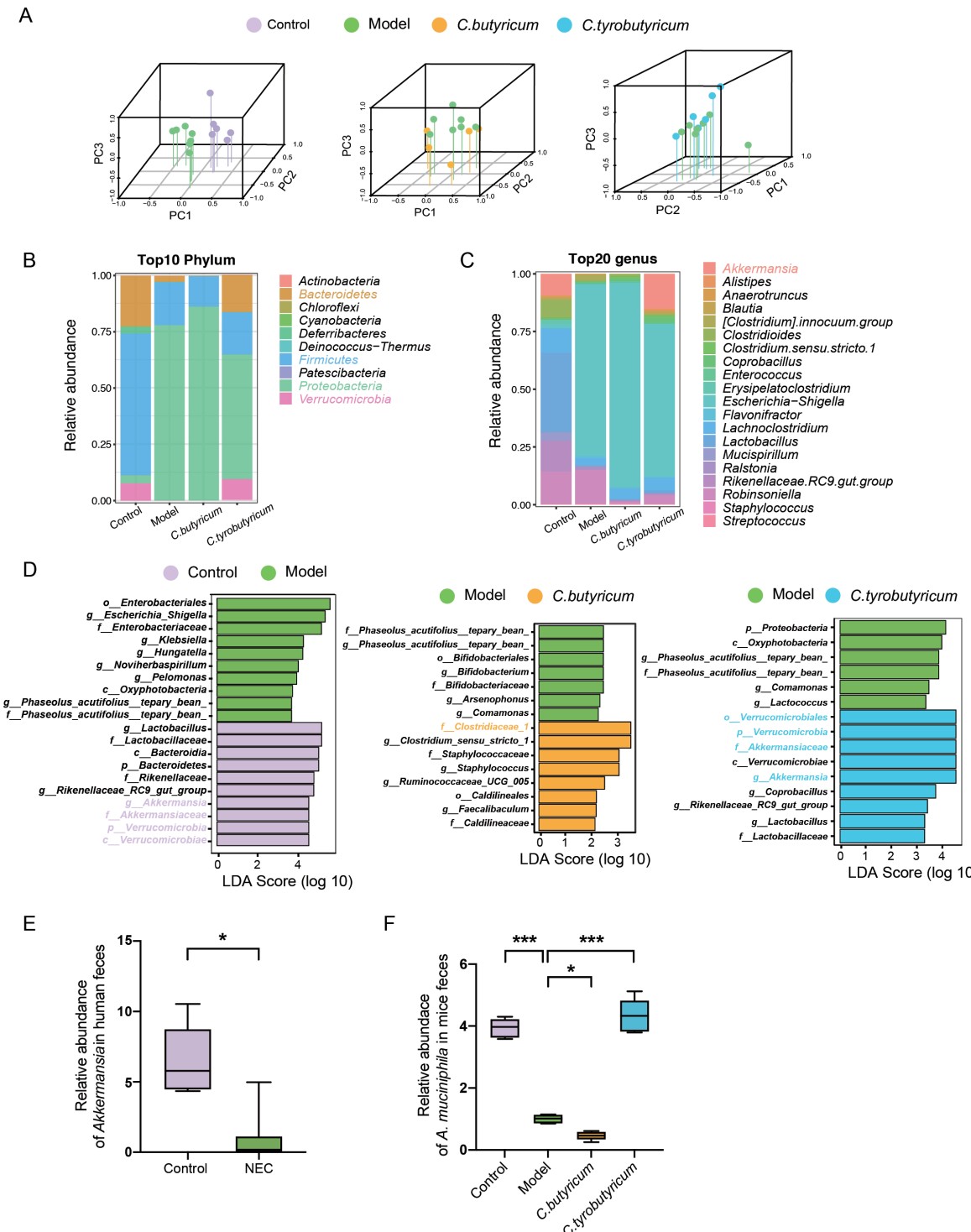

**FIG 5** The positive effect of *C. tyrobutyricum* and the negative effect of *C. butyricum* on NEC were associated with modulating the level of *A. muciniphila*. (A) Representative images of three-dimensional principal component analysis (PCA) of control, model, *C. butyricum* and *C. tyrobutyricum* mice (*n* = 5–7). (B and C) The relative abundances of bacteria at phylum and genus level were shown as a stacked bar plot. Each column corresponds to one group (*n* = 5–7). (D) Representative images of linear discriminant analysis effect size (LEfSe) of each two groups (*n* = 5–7). (E) Relative abundance of *Akkermansia* in neonate feces analyzed by Kruskal-Wallis test (*n* = 6–7). (F) Relative 16S rDNA expression of *A. muciniphila* measured by quantitative PCR analyzed by one-way ANOVA (*n* = 5). Quantified results were represented using box and whisker plots. p-Values were generated by one-way ANOVA with multiple comparisons. ***$P < 0.001$.

*tyrobutyricum* treatment reduced inflammation and improved intestinal barrier integrity, while *C. butyricum* treatment exacerbated the condition, according to our data. We also investigated the bacterial diversity and homeostasis of human and mouse feces to identify potential pathogenic mechanisms. The 16S rDNA sequencing results indicated that the decline of *A. muciniphila* was associated with NEC development. Our findings supported the idea that *C. tyrobutyricum* treatment restored the level of *A. muciniphila*, while *C. butyricum* treatment exacerbated the loss of *A. muciniphila*. In addition, the fitness advantage of *C. butyricum* in interspecific competition may account for the reduction in *A. muciniphila*.

Although our study provided clear evidence of the roles of *C. butyricum* and *C. tyrobutyricum* in NEC progression, there are still some limitations to our study that need to be addressed. For example, we did not quantify the species level of *C. butyricum* and *C. tyrobutyricum* in mouse feces to provide a more detailed description of the microbiota composition *in vivo*. Additionally, the negative result of the competition between *C. tyrobutyricum* and *A. muciniphila* suggests another potential mechanism for exploring the fitness advantage of *A. muciniphila* under *C. tyrobutyricum* treatment.

In conclusion (Fig. 7), our study strongly supports the idea that *C. tyrobutyricum* treatment, but not *C. butyricum* treatment, is suitable for protecting against NEC by reducing intestinal inflammation and improving intestinal barrier integrity. Moreover, the

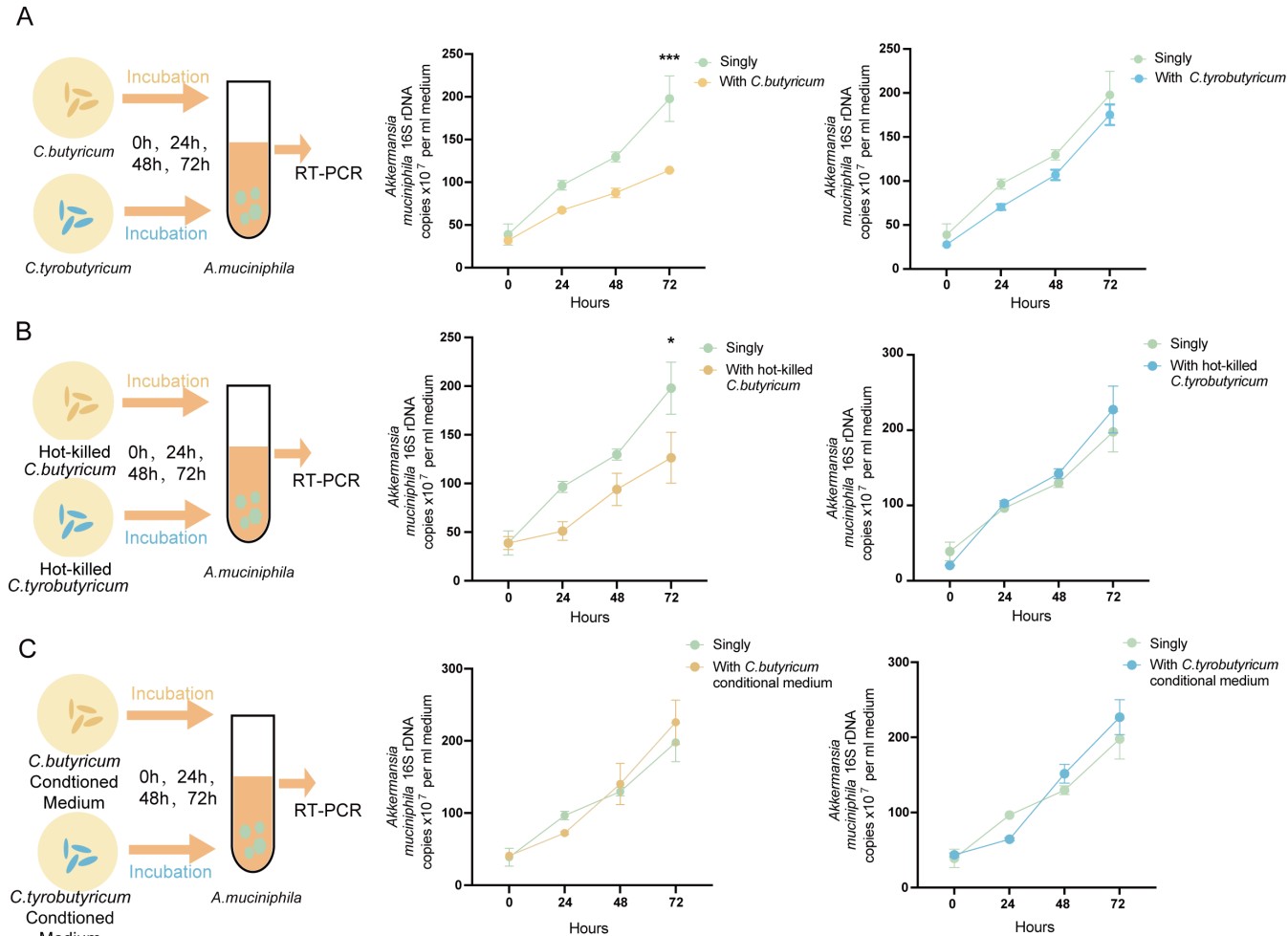

**FIG 6** Interspecific competition provided a fitness advantage to *C. butyricum* over *A. muciniphila*. (A–C) Schematic diagram demonstrating the workflow of competition experiments in liquid cultures among different strains and the growth rates of *A. muciniphila* in each co-culture solution were represented by line charts. The growth rates of *A. muciniphila* were shown as the 16S rDNA copies measured by quantitative PCR (*n* = 3). Quantified results were shown as mean ± SEM. *P*-values were generated by two-way ANOVA with multiple comparisons. *$P < 0.05$, **$P < 0.01$,*** $P < 0.001$.

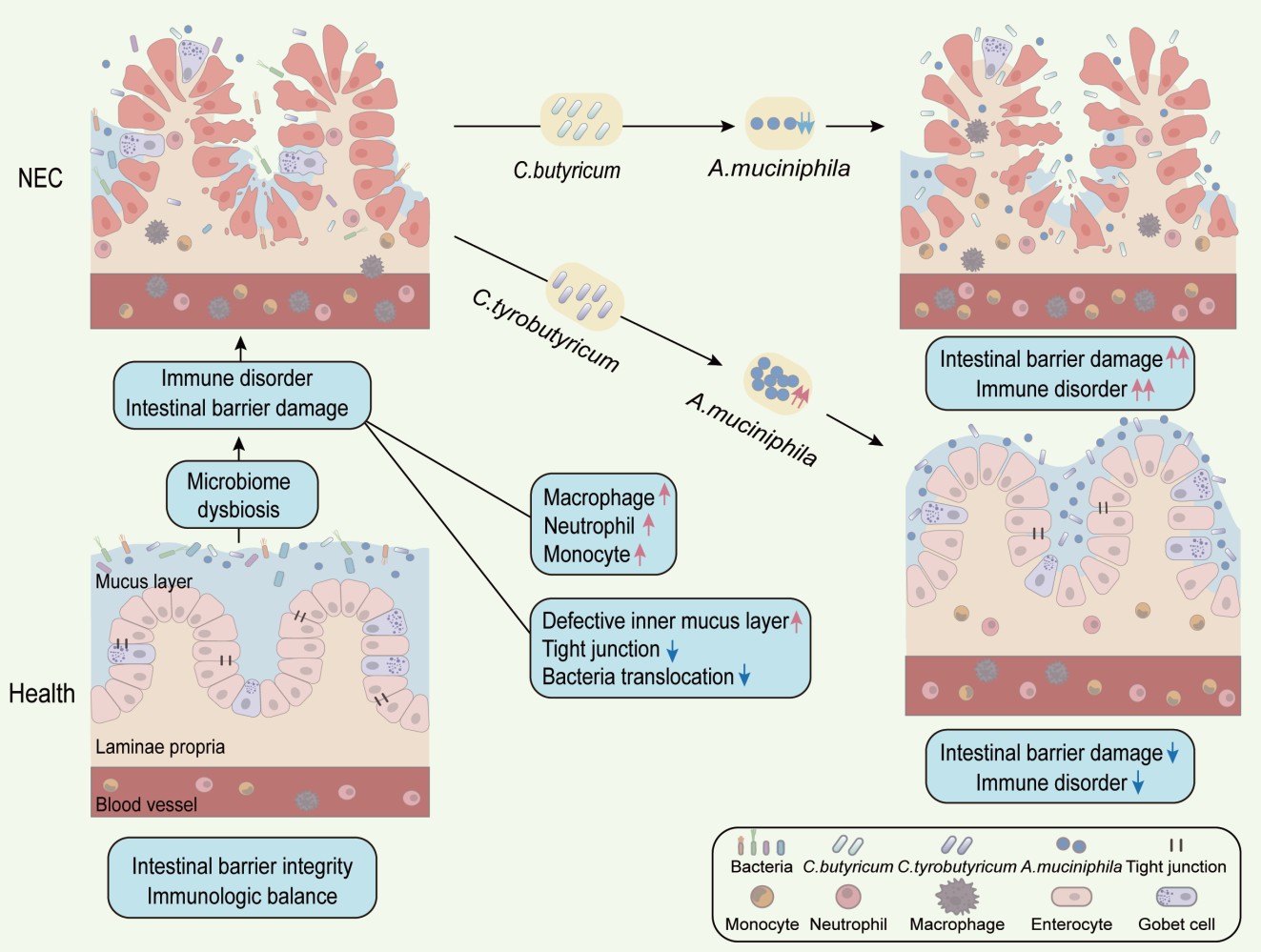

**FIG 7** *C. butyricum* and *C. tyrobutyricum* affect the development of NEC by regulating the relative abundance of *A. muciniphila*.

two types of treatment have opposite effects on modulating the level of *A. muciniphila*, which is closely associated with NEC development. Therefore, *C. tyrobutyricum* supplementation may have the potential to become a therapeutic strategy for NEC in clinical treatment.

## ACKNOWLEDGMENTS

Thanks are due to Jiangning affiliated Hospital of Nanjing Medical University and Nanjing Children's Hospital for assistance with the project. The authors thank Nanjing Jiangbei New Area Biopharmaceutical Public Service Platform for analyzing the sequencing data. We appreciate the instruction and help from Prof. Lu and Dr. Wei, our mentors in life and research.

This work was financially supported by Natural Science Foundation of Jiangsu Province (BK20200154) and National Natural Science Foundation of China (82004124). This project was supported in part by the Open Project of Chinese Materia Medica First-Class Discipline of Nanjing University of Chinese Medicine (2020YLXK20) and the Open Project of State Key Laboratory of Natural Medicines (SKLNMKF202301).

Z.W. and Y.L. designed the study protocol, supervised all parts of the project, and reviewed final version approval. W.T. contributed to the design of the clinical trial and the analysis of clinical data. R.T. and G.Z. performed the experiments and analyzed the data. H.L. and S.X. were responsible for the collection and analysis of clinical samples. Y.P.,

P.C., and R.D. contributed reagents/materials/analysis tools. R.T., G.Z., and Z.W. edited the manuscript. W.C. and A.W. checked data and provided suggestions. Y.L. contributed to text revision and discussion. All authors read and approved the final manuscript.

## AUTHOR AFFILIATIONS

[1]Jiangsu Key Laboratory for Pharmacology and Safety Evaluation of Chinese Materia Medica, School of Pharmacy, Nanjing University of Chinese Medicine, Nanjing, China
[2]Department of Pediatric Surgery, Children's Hospital of Nanjing Medical University, Nanjing, China
[3]Jiangsu Joint International Research Laboratory of Chinese Medicine and Regenerative Medicine, Nanjing University of Chinese Medicine, Nanjing, China
[4]Ningbo Women and Children's Hospital, Ningbo, China

## AUTHOR ORCIDs

Weibing Tang http://orcid.org/0000-0002-3582-3408
Yin Lu http://orcid.org/0000-0003-2063-8485
Zhonghong Wei http://orcid.org/0000-0002-1733-6331

## FUNDING

| Funder | Grant(s) | Author(s) |
| --- | --- | --- |
| Natural Science Foundation of Jiangsu Province (Jiangsu Natural Science Foundation) | BK20200154 | Zhonghong Wei |
| National Natural Science Foundation of China | 82004124 | Zhonghong Wei |
| The Open Project of State Key Laboratory of Natural Medicines | SKLNMKF202301 | Zhonghong Wei |

## DATA AVAILABILITY

Raw 16S rRNA sequencing data have been deposited in Sequence Read Archive with BioProject ID PRJNA894504. The other data are available from the corresponding author upon reasonable request. All data needed to evaluate the conclusions in the paper are present in the paper. Additional data related to this paper may be requested from the authors.

## ETHICS APPROVAL

All experimental protocols were approved by the Animal Care and Use Committee of the Nanjing University of Chinese Medicine (Nanjing, China) and conformed to the Guidelines for the Care and Use of Laboratory Animals (I ACUC-1908022, 5 August 2019). The study had been approved by the Medical Ethics Committee of the Children's Hospital of Nanjing Medical University (202101014-1).

## ADDITIONAL FILES

The following material is available online.

### Supplemental Material

**Fig. S1 (mSystems00732-23-s0001.tif).** Analysis of microbial networks.
**Fig. S2 (mSystems00732-23-s0002.tif).** Alleviation or aggravation of intestinal inflammation.
**Fig. S3 (mSystems00732-23-s0003.tif).** Protection or disruption of intestinal barrier integrity.
**Fig. S4 (mSystems00732-23-s0004.tif).** Effects on NEC.

**Supplemental legends (mSystems00732-23-s0005.docx).** Legends to Table S1 and Fig. S1 to S4.

**Table S1 (mSystems00732-23-s0006.xlsx).** Information on human samples.

## Open Peer Review

**PEER REVIEW HISTORY (review-history.pdf).** An accounting of the reviewer comments and feedback.

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
