## [Reviewer comments · mSystems]

Clostridium butyricum and Clostridium tyrobutyricum: angel or devil for Necrotizing enterocolitis?

Zhong Wei, Ruizhi Tao, Gang Zong, Ye Pan, Hong Li, Peng Cheng, Rui Deng, Wenxing Chen, Aiyun Wang, Shi Xia, Wen Tang, and Yin Lu

Corresponding Author(s): Zhong Wei, Nanjing University of Chinese Medicine

Review Timeline:

Submission Date:	August 8, 2023
Editorial Decision:	August 17, 2023
Revision Received:	August 21, 2023
Accepted:	September 6, 2023

Editor: Hongwei Zhou

Reviewer(s): Disclosure of reviewer identity is with reference to reviewer comments included in decision letter(s). The following individuals involved in review of your submission have agreed to reveal their identity: Yunlong Shan (Reviewer #5)

Transaction Report:

DOI: <https://doi.org/10.1128/msystems.00732-23>

August 17, 2023

Dr. Zhong hong Wei
Nanjing University of Chinese Medicine
Nanjing
China

Re: mSystems00732-23 (Clostridium butyricum and Clostridium tyrobutyricum: angel or devil for Necrotizing enterocolitis?)

Dear Dr. Zhong hong Wei:

Thank you for submitting your manuscript to mSystems. We have completed our review and I am pleased to inform you that, in principle, we expect to accept it for publication in mSystems. However, acceptance will not be final until you have adequately addressed the reviewer comments.

Preparing Revision Guidelines

Please return the manuscript within 60 days; if you cannot complete the modification within this time period, please contact me. If you do not wish to modify the manuscript and prefer to submit it to another journal, please notify me of your decision immediately so that the manuscript may be formally withdrawn from consideration by mSystems.

Sincerely,

Hongwei Zhou

Editor, mSystems

Journals Department
Reviewer comments:

Reviewer #1 (Comments for the Author):

The author has addressed all my concern.

Reviewer #3 (Comments for the Author):

The study offers valuable insights into the pathogenesis of Necrotizing enterocolitis and the opposing effects of *Clostridium butyricum* and *Clostridium tyrobutyricum*. The findings have promising implications for potential interventions for the disease. It is intriguing to see the contrasting effects of the two *Clostridia* on the intestinal barrier and inflammation.

Furthermore, the study provides evidence that changes in *A. muciniphila* levels associated with the two *Clostridia* can either exacerbate or alleviate the disease in both in vivo and in vitro models, shedding light on the complex interactions between these microorganisms.

However, there are still some points that need to be modified or supplemented:

1. The manuscript has some repetition in the content of the conclusion section, which the authors should address.
2. The colors and font size of all the bar charts should be consistent, and the scar bar should be highlighted."
3. The authors have provided comprehensive 16S rDNA data results. It is recommended that they also provide Picrust2 predicted data to uncover the potential role of microorganisms in disease development.

The study offers valuable insights into the pathogenesis of Necrotizing enterocolitis and the opposing effects of *Clostridium butyricum* and *Clostridium tyrobutyricum*. The findings have promising implications for potential interventions for the disease. It is intriguing to see the contrasting effects of the two *Clostridia* on the intestinal barrier and inflammation.

Furthermore, the study provides evidence that changes in *A. muciniphila* levels associated with the two *Clostridia* can either exacerbate or alleviate the disease in both in vivo and in vitro models, shedding light on the complex interactions between these microorganisms.

However, there are still some points that need to be modified or supplemented:

1. The manuscript has some repetition in the content of the conclusion section, which the authors should address.
2. The colors and font size of all the bar charts should be consistent, and the scar bar should be highlighted."
3. The authors have provided comprehensive 16S rDNA data results. It is recommended that they also provide Picrust2 predicted data to uncover the potential role of microorganisms in disease development.

Reviewer #1

Thank you for your carefully consideration and review. Those following are the point-by-point responses to the issues:

1. Please describe in detail the source of control group samples for transcriptome sequencing.

Response:

This is an excellent question. Firstly, it is not ethical to collect colon tissues from healthy infants, so we collected colon tissues from infants with non-inflammatory diseases (specifically, those with megacolon stenosis) as our control group. We then compared the results of transcriptome sequencing between the two groups in order to investigate the differences in gene expression and gain insights into the underlying mechanisms of the disease.

2. Please describe in detail the source of control group samples for 16S rDNA sequencing.

Response

Thank you for your question. To serve as our control group, we collected 10 fecal samples from healthy infants at the Jiangning affiliated Hospital of Nanjing Medical University. We then used 16S rDNA sequencing to compare the microbial composition between our control group and the group of NEC.

3. Regarding the last result, please discuss in the conclusion the possible reasons why *C. butyricum* did not inhibit the growth of *A. muciniphila* in vitro.

Response:

Thank you for your question, it is very interesting. If I understand correctly, you are

asking why in vitro experiments with *C.tyrobutyricum* do not promote the growth of *Akkermansia muciniphila*. Our results show that *C. butyricum* can inhibit the growth of *Akkermansia muciniphila* in vitro co-culture experiments, and even heat-killed *C. butyricum* can suppress the growth of *A muciniphila*. This suggests that *C. butyricum* themselves may be inhibiting the growth of *A muciniphila* directly. This result is consistent with our 16S rDNA sequencing results. However, on the contrary, we have not detected any promotion of *A muciniphila* growth by *C.tyrobutyricum*, and the underlying reasons for this may be complex. One possibility is that the limited energy in the co-culture system may not fully simulate the growth environment of the two bacteria in the intestine. Another possibility is that the promoting/ inhibiting effect of *C.tyrobutyricum* on *A muciniphila* growth may be due to the indirect effect of a third type of bacteria, in which the supplementation of *C.tyrobutyricum* promotes/inhibits the abundance of the third type of bacteria, indirectly promoting the colonization and growth of *A muciniphila*. Based on our 16S rDNA sequencing results, there may be specific species of *Proteobacteria* or *Lactobacillaceae* that are worth considering in relation to our research question.

Reviewer #2

Thank you very much for your valuable comments and review. Those following are the point-by-point responses to all the issues:

1. As regarding to experimental design for "Gut microbiota took part in the development of NEC" shown in figure 1. Authors should add the experiment transplant feces from NEC patients to germfree mice to check whether feces from NEC patients can induce NEC-related symptom. The current animal model with hypoxia inducement cannot clarify the feces from NES affect development of NEC. Hypoxia would induce huge microbiota change.

Response:

Thank you for your valuable feedback. I think this is a very meaningful question. It has been established that changes in the gut microbiota occur prior to the onset of NEC, but gut dysbiosis is not the only factor influencing NEC. In other words, the development of NEC is a multifactorial process. Therefore, we used a hypoxia combined with cold stimulation modeling method and evaluated the effects of human fecal samples and different *Clostridium* strains on the progression of NEC through gavage. Our results also showed that administering human fecal samples to NEC model mice promoted the progression of NEC. Of course, exploring the relationship between the gut microbiota and the development of NEC more directly through germ-free mice is also a good suggestion and starting point. However, due to experimental constraints and logical considerations, we did not consider using germ-free mice before starting the experiment.

Reference:

JACOB J A. In Infants With Necrotizing Enterocolitis, Gut Dysbiosis Precedes Disease [J]. JAMA, 2016, 315(21): 2264-5.<https://doi.org/10.1001/jama.2016.4341>.

NINO D F, SODHI C P, HACKAM D J. Necrotizing enterocolitis: new insights into pathogenesis and mechanisms [J]. Nat Rev Gastroenterol Hepatol, 2016, 13(10): 590-600.<https://doi.org/10.1038/nrgastro.2016.119>.

2. What is scientific logical relationship between Figure 1 and Figure2? According to Figure1, we did see genus, Clostridia have significant change between healthy group and NEC group. Both *C.tyrobutyricum* or *C.butyricum* were isolated from your NEC patients? I didn't see any description about the source of both strains. Authors need check whether both strains have change in human samples through PCR?

Response:

Thank you for your valuable feedback. This question is also very important. It has been reported that there is significant dysbiosis of the *Clostridium* genus in the pathogenesis of NEC. Some studies have shown significant changes in *C. butyricum*,

but there are few reports on *C. tyrobutyricum*. However, both of these strains have been widely used in the field of probiotics. Therefore, to explore the roles of these two bacteria in the NEC process and whether they could serve as potential probiotic preparations for the treatment of NEC patients, we conducted an in-depth investigation. We obtained these two strains through purchase and conducted subsequent studies. We also considered using PCR to detect changes in these two strains in the human samples we collected. Unfortunately, due to the numerous strains in the Clostridium genus, we could not obtain specific primer sequences that could accurately detect only *C. butyricum* or *C. tyrobutyricum* in fecal samples. Also, due to experimental limitations at the time, we only performed 16S rDNA sequencing and did not perform metagenomic analysis.

Reference:

CASSIR N, BENAMAR S, LA SCOLA B. Clostridium butyricum: from beneficial to a new emerging pathogen [J]. Clin Microbiol Infect, 2016, 22(1): 37-45. <https://doi.org/10.1016/j.cmi.2015.10.014>.

SIM K, SHAW A G, RANDELL P, et al. Dysbiosis anticipating necrotizing enterocolitis in very premature infants [J]. Clin Infect Dis, 2015, 60(3): 389-97. <https://doi.org/10.1093/cid/ciu822>.

3. Accumulating evidence suggests that disruption of the T helper 17 (Th17) cell and FoxP3+ regulatory T (Treg) cell balance is an important factor underlying the powerful inflammatory response in NEC. Why did authors only check macrophages, monocytes and neutrophils in the intestinal lamina propria immune cells through flow cytometric analysis? Authors should present reason. How about Th17/Treg cells?

Response:

Thank you for your valuable feedback. As the disruption of the T helper 17 (Th17) cell and FoxP3⁺ regulatory T (Treg) cell balance is known to play an important role in NEC, we conducted a flow cytometry analysis to compare the Th17/Treg cells in four different groups. Our results showed that the number of CD4⁺ RORγt⁺ Th17 cells and CD25⁺ FoxP3⁺ Treg cells was significantly altered in both the colon and ileum. Specifically, the NEC model disrupted the Th17/Treg cell balance in the ileum and colon. Treatment with *C.tyrobutyricum* reversed the decrease of Th17 cells in NEC model mice and decreased the number of Treg cells, while *C.butyricum* failed to produce the same effect.

Other comments:

1. The manuscript has been edited and refined to improve the precision of its content.

If there are any remaining grammar or spelling errors, please let us know.

2. We have included the immunofluorescence results of the colon in the supplementary materials (Figure S3A). These results showed that the tight junction (ZO-1, E-cadherin, and Claudin-1) of the colon tissue was damaged in the NEC model. Treatment with *C.tyrobutyricum* alleviated the damage, while treatment with *C.butyricum* worsened it.

3. The number of mice is noted in all legends of animal experiment to ensure the rigor of our results

4. All the magnification times are noted in the legends of IF pictures.

5. The analysis of microbial networks in infants' feces has been included in the supplementary materials. Based on the results, it suggests potential interactions between specific groups of gut microbiota.

6. The ethics approval No. of human subjects has been added to the part of Methods, the study had been approved by the Medical Ethics Committee of the Children's Hospital of Nanjing Medical University (202101014-1).

Reviewer #3

Thank you very much for your valuable comments and review. Those following are the point-by-point responses to all the issues:

1.The manuscript has some repetition in the content of the conclusion section, which the authors should address.

Response:

Thank you for your attention to detail. We have carefully reviewed the content and removed any instances of repetition in the conclusion section. If there are any lingering grammar or spelling errors, we would appreciate your feedback and assistance in identifying them.

2.The colors and font size of all the bar charts should be consistent, and the scale bar should be highlighted.

Response:

Thank you for your suggestion. Firstly, all the colors in each figure have been standardized and we have ensured consistent font sizes across all the pictures, enhancing readability and visual consistency throughout. Furthermore, the scale bar has been appropriately highlighted.

3.The authors have provided comprehensive 16S rDNA data results. It is recommended that they also provide Picrust2 predicted data to uncover the potential role of microorganisms in disease development.

Response:

Thank you for your advice. The results from PICRUST2 prediction revealed that the distinct microbiome in NEC patients significantly influenced the metabolism and biosynthesis of amino acids, particularly arginine, proline, and histidine, which are essential for the host. It's important to note that these predictions were based on the KEGG database.

September 6, 2023

Dr. Zhong hong Wei
Nanjing University of Chinese Medicine
Nanjing
China

Re: mSystems00732-23R1 (Clostridium butyricum and Clostridium tyrobutyricum: angel or devil for Necrotizing enterocolitis?)

Dear Dr. Zhong hong Wei:

Your manuscript has been accepted, and I am forwarding it to the ASM Journals Department for publication. For your reference, ASM Journals' address is given below. Before it can be scheduled for publication, your manuscript will be checked by the mSystems production staff to make sure that all elements meet the technical requirements for publication. They will contact you if anything needs to be revised before copyediting and production can begin. Otherwise, you will be notified when your proofs are ready to be viewed.

If you would like to submit a potential Featured Image, please email a file and a short legend to msystems@asmusa.org. Please note that we can only consider images that (i) the authors created or own and (ii) have not been previously published. By submitting, you agree that the image can be used under the same terms as the published article. File requirements: square dimensions (4" x 4"), 300 dpi resolution, RGB colorspace, TIF file format.

We recognize that the video files can become quite large, and so to avoid quality loss ASM suggests sending the video file via <https://www.wetransfer.com/>. When you have a final version of the video and the still ready to share, please send it to mSystems staff at msystems@asmusa.org.

Sincerely,

Hongwei Zhou
Editor, mSystems

Journals Department
E-mail: mSystems@asmusa.org